# Introducing carbon assimilation in yeasts using photosynthetic directed endosymbiosis

Yang-le Gao [1], Jay Cournoyer [1,6], Bidhan C. De[1,6], Catherine L. Wallace [2], Alexander V. Ulanov[3], Michael R. La Frano[3] & Angad P. Mehta [1,4,5] ✉

Conversion of heterotrophic organisms into partially or completely autotrophic organisms is primarily accomplished by extensive metabolic engineering and laboratory evolution efforts that channel $CO_2$ into central carbon metabolism. Here, we develop a directed endosymbiosis approach to introduce carbon assimilation in budding yeasts. Particularly, we engineer carbon assimilating and sugar-secreting photosynthetic cyanobacterial endosymbionts within the yeast cells, which results in the generation of yeast/cyanobacteria chimeras that propagate under photosynthetic conditions in the presence of $CO_2$ and in the absence of feedstock carbon sources like glucose or glycerol. We demonstrate that the yeast/cyanobacteria chimera can be engineered to biosynthesize natural products under the photosynthetic conditions. Additionally, we expand our directed endosymbiosis approach to standard laboratory strains of yeasts, which transforms them into photosynthetic yeast/cyanobacteria chimeras. We anticipate that our studies will have significant implications for sustainable biotechnology, synthetic biology, and experimentally studying the evolutionary adaptation of an additional organelle in yeast.

Biotechnology platforms rely on genetically tractable, model organisms for producing fine chemicals and proteins. Typically, these model organisms (bacteria like *Escherichia coli* and yeasts like *Saccharomyces cerevisiae*) are heterotrophic organisms that rely on exogenous carbon feedstocks like glycerol or glucose. Efforts have been made to transform heterotrophic model yeasts like *S. cerevisiae* and *Pichia pastoris*, and model bacteria like *Escherichia coli* and *Methylobacterium extorquens*, into partially or completely autotrophic organisms that are capable of assimilating carbon from $CO_2$ as a feedstock carbon source[1]. All the studies rely on extensive metabolic engineering, which is often followed by adaptive evolution experiments that result in the channeling of $CO_2$ into the central carbon metabolism. Heterologous expression of the components of the naturally existing $CO_2$-fixation pathways like the Calvin–Benson–Bassham (CBB) cycle, along with the expression of accessory proteins, is typically used for these metabolic engineering efforts[1]. For example, expressing RuBisCO and phosphoribulokinase (PRK) in *E. coli* led to an increased in situ $CO_2$ recycling in the engineered strains[2]. Extensive metabolic engineering and laboratory evolution experiments resulted in the generation of hemiautotrophic and autotrophic *E. coli* strains that possess a functional CBB cycle. Similarly, metabolic engineering and adaptive evolution efforts have been performed in yeasts, *P. pastoris*, to engineer strains

[1]Department of Chemistry, University of Illinois at Urbana-Champaign, 600 S Mathews Avenue, Urbana, Illinois, US. [2]The Imaging Technology Group, Beckman Institute for Advanced Science & Technology, University of Illinois at Urbana-Champaign, 405 North Mathews Avenue, Urbana, IL, US. [3]Carver Metabolomics Core, Roy J. Carver Biotechnology Center, University of Illinois at Urbana-Champaign, 1206 West Gregory Drive, Urbana, Illinois, US. [4]Carl R. Woese Institute for Genomic Biology, University of Illinois at Urbana-Champaign, 1206 West Gregory Drive, Urbana, Illinois, US. [5]Cancer Center at Illinois, University of Illinois at Urbana-Champaign, 405 North Mathews Avenue, Urbana, IL, US. [6]These authors contributed equally: Jay Cournoyer, Bidhan C. De. ✉e-mail: apm8@illinois.edu

that were capable of growing on $CO_2$ as a feedstock carbon source[3]. Additionally, components of the CBB cycle have also been introduced in *S. cerevisiae* for carbon fixation and improving ethanol yields[4–6]. Metabolic engineering efforts have also been performed in autotrophs to produce molecules like ethylene, pyruvate, butanediol amongst others[7–9]. Photomixotrophic chemical production has also been explored in cyanobacteria and microalgae[10,11]. Further, microfluidic-based approaches have been used to generate cell-sized droplets that are able to perform light-powered $CO_2$ fixation[12]. Engineering carbon assimilation pathways in model organisms as well as synthetic systems are expected to have a significant impact on various biotechnology applications, including bioproduction of proteins and other chemicals.

Our approach for engineering carbon assimilation was inspired by the endosymbiotic theory of chloroplast evolution, which suggests that the modern-day chloroplasts evolved from once free-living cyanobacteria that were established as endosymbionts within the host cells. These endosymbiotic events resulted in a chimeric cell-within-cell system (endosymbionts within a host cell) of two or more organisms which eventually transformed eukaryotic cells into modern-day photosynthetic cells like algal cells and land plant cells containing intracellular plastids[13–15]. It is suggested that the cyanobacterial photosynthesis may have been one of the key drivers of endosymbiosis and chloroplast evolution[16,17]. Over a period of time, the cyanobacterial endosymbiont/chloroplast evolved to perform a variety of functions for the host cells, including photosynthetic carbon assimilation, photophosphorylation to generate ATP along with several metabolic functions like amino acid and fatty acid biosynthesis, sulfate assimilation and nitrate assimilation[18]. Inspired by these observations, we have launched an area of investigation, i.e., directed endosymbiosis, where we artificially engineer endosymbiosis between genetically tractable model organisms for evolutionary studies and synthetic biology[19–21]. Particularly, we have been developing directed endosymbiosis between cyanobacteria and yeast cells for: (i) studying the evolutionary adaptation of an additional photosynthetic endosymbiont/organelle in yeast, and (ii) exploiting this endosymbiont/host chimeras for synthetic biology (e.g., endosymbiont/organelle mediated conversion of $CO_2$ to chemicals)[19,22].

Here, we investigate if we can introduce carbon assimilation in yeasts by engineering directed endosymbiosis between yeast and photosynthetic cyanobacteria. We engineer various mutants of cyanobacteria as endosymbionts within model yeasts, *S. cerevisiae*, such that the endosymbiotic cyanobacteria perform carbon assimilation and provide assimilated sugars along with photosynthetically generated ATP to the mutant yeast cells. Several of the resulting yeast/cyanobacteria endosymbiotic chimeras are able to propagate for several generations in the presence of bicarbonate as a carbon source and in the absence of feedstock carbon sources like glucose or glycerol. Further, as a proof-of-concept experiment, we demonstrate that orthogonal metabolic pathways can be engineered in the yeast/cyanobacteria chimeras to biosynthesize natural products under photosynthetic conditions. Additionally, we develop relatively simple approaches that utilize standard yeast genetics and directed endosymbiosis to introduce cyanobacterial endosymbionts within standard laboratory strains of yeast. We believe that such approaches could open up frontiers in biotechnology and sustainable synthetic biology[23]. These platforms can also be used to experimentally study the evolutionary adaptation of an additional endosymbiont/organelle in yeast and obtain molecular insights into the transformation of bacterial endosymbionts into organelles.

## Results

### Engineering Syn7942 to secrete glucose assimilated from photosynthesis

For our studies, we used *Synechococcus elongatus* PCC 7942 (Syn7942), a genetically tractable model cyanobacterium, as the parent strain[24,25].

Our first goal was to engineer Syn7942 mutants to secrete photosynthetically assimilated sugars as glucose. To accomplish this, we performed metabolic engineering in Syn7942 such that the engineered strains were able to break down sucrose into glucose and fructose and were further able to secrete glucose (Fig. 1A)[26]. We began by constructing integrative plasmids that constitutively expressed codon-optimized genes corresponding to *glf* and *invA* gene from *Zymomonas mobilis* (Fig. 1B). The *glf* gene encodes for a glucose facilitator gene that acts as a glucose transporter that is able to secrete glucose upon build-up and the *invA* gene encodes for an intracellular invertase that breaks down sucrose into glucose and fructose[26–28]. Starting with pCV0063, we constructed a plasmid (pYG10) encoding: (i) codon-optimized *glf* gene, (ii) codon-optimized *invA* gene, (ii) a spectinomycin resistance cassette and (iii) NSI homology region for genomic recombination (Supplementary Fig. 1). Similarly, we made a construct expressing only the *glf* gene (pYG11) and used pCV0063 as a control lacking expression of both *glf* and *invA* (pCV0063). We transformed Syn7942 with pYG10, pYG11 and pCV0063 and selected on BG-11 medium plates containing spectinomycin (2 µg/mL) and streptomycin (2 µg/mL) to generate the strain SynYLG3, SynYLG4 and SynYLG5 respectively. Single colonies were selected, and grown in liquid BG-11 medium containing spectinomycin (2 µg/mL) and streptomycin (2 µg/mL), the genomic DNA was isolated, and the presence of insertion at the NSI site was confirmed by locus-specific PCR analysis (Supplementary Fig. 2). Next, the cells were cultured under photosynthetic sucrose accumulation condition under which the assimilated sucrose is expected to breakdown into glucose and fructose due to the enzymatic activity of InvA and the glucose is expected to be secreted because of the expression of Glf protein. We analyzed the extracellular culture medium of SynYLG3, SynYLG4 and SynYLG5 for the presence of glucose using colorimetric assays for glucose detection (Glucose Assay Kit (GAGO20-1KT, Sigma, USA). As demonstrated in Fig. 1C, no glucose is detected in the extracellular culture medium for the control mutants lacking *glf* and *invA*. On the other hand, low levels of glucose are detected in the extracellular culture medium for cells expressing Glf protein alone, and much higher levels of glucose are detected in the extracellular culture medium for cells expressing both Glf and InvA proteins.

### Engineering cyanobacteria mutants to incorporate genetic elements necessary for endosymbiosis

Our next goal was to incorporate genetic elements in SynYLG3 to transform it into yeast endosymbionts. We had previously demonstrated that we can engineer endosymbiosis between cyanobacterial mutants and yeast mitochondrial mutant, *S. cerevisiae cox2-60*[19]. Briefly, the host yeast strain *S. cerevisiae cox2-60* is incapable of assembling a functional cytochrome *c* oxidase complex and consequently has a respiration-deficient phenotype specifically due to the lack of ATP synthesis under the defined selection conditions. Using directed endosymbiosis, we introduce cyanobacterial mutants within *S. cerevisiae cox2-60* cells where under the defined photosynthetic selection conditions, the cyanobacterial endosymbionts provide photosynthetically generated ATP to *S. cerevisiae cox2-60* and the *S. cerevisiae cox2-60* provide defined metabolites (e.g., methionine) to the cyanobacterial endosymbionts. For this system, the optimal cyanobacterial mutant (SynJEC3) expressed ADP/ATP translocase Ntt1, from *Protochlamydia amoebophila* UWE25, and SNARE-like proteins IncA and CT813 from *Chlamydia trachomatis*[19]. In this previous platform, the endosymbionts were not engineered to provide assimilated carbon sources to the yeast cells. SynJEC3 mutant strain was generated by the transformation of Syn7942 with pML17, which is an integration vector with a NSII integration site on the cyanobacterial genome. This integration cassette contains *ntt1*, *incA* and *CT_813* and chloramphenicol selection marker. Because we wanted to further engineer SynYLG3 as an endosymbiont within *S. cerevisiae cox2-60*, we

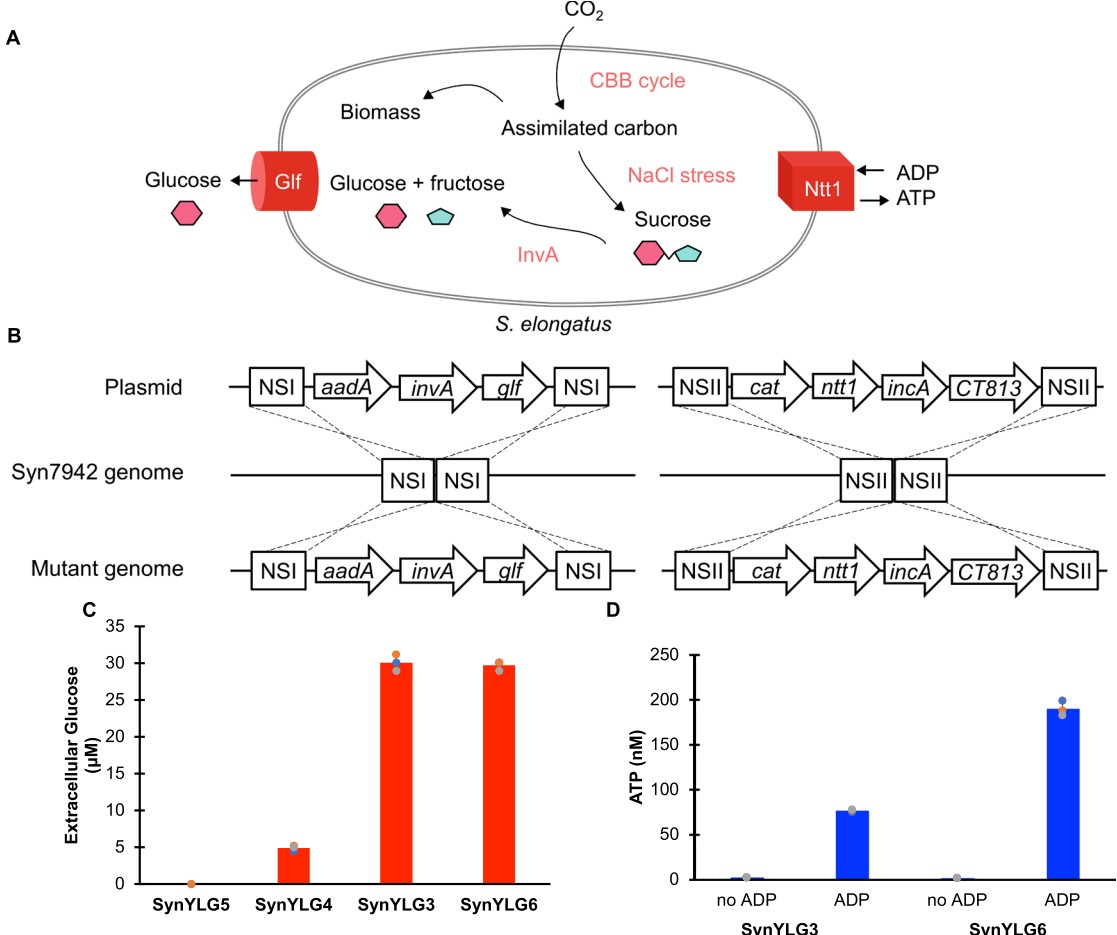

**Fig. 1 | Engineering Syn7942 to secrete glucose assimilated from photosynthesis. A** Engineering Syn7942 to secrete glucose assimilated from $CO_2$. **B** Suicide plasmid-based strategy used in this manuscript to engineer cyanobacterial mutants, SynYLG strains. **C** Secretion of extracellular glucose by SynYLG6 and SynYLG3 cells expressing the *glf* and *invA* genes in comparison to SynYLG4 cells only expressing the *glf* gene and SynYLG5 cells (no *glf* and *invA*). **D** Release of ATP by *SynYLG6* cells expressing the *UWE25* ADP/ATP translocase in the presence of 80 μM ADP in comparison to SynYLG3 cells. As compared to SynYLG3 cells (no ATP/ADP translocase), higher levels of ATP was released when SynYLG6 (expressing the ATP/ADP translocase) were challenged with extracellular ADP (80 μM). But no ATP was released when the cells were challenged with a blank solution lacking ADP ($n$ = 3 biological replicates; data are presented as mean values + /− SEM). Source data are provided as a Source Data file.

transformed SynYLG3 with pML17 and selected the transformants in BG-11 medium containing spectinomycin (2 μg/mL), streptomycin (2 μg/mL) and chloramphenicol (7.5 μg/mL). Individual colonies were cultured in a liquid medium and their genomes were analyzed for recombination. As expected, we detected the presence of an integrated DNA fragment derived from pML17 at the NSII site in SynYLG3 (Supplementary Fig. 3). This mutant cyanobacterial strain was called SynYLG6. We cultured SynYLG6 in BG-11 medium containing spectinomycin (2 μg/mL), streptomycin (2 μg/mL) and chloramphenicol (7.5 μg/mL). We then analyzed the extracellular culture medium for the presence of glucose, as described before. We detected similar levels of glucose in the extracellular culture medium of SynYLG6 as compared to SynYLG3 (Fig. 1C). Similarly, we performed luciferase assays to determine the ADP/ATP translocase activity[19,20]. We observed that the SynYLG6 strain released a higher amount of ATP as compared to background levels of ATP secreted by SynYLG3 strains when challenged with extracellular ADP. On the other hand, when ADP was not added, the strains did not secrete ATP (Fig. 1D). The SynYLG3 strain releases background levels of ATP upon ADP challenge, this level is similar to the amount of ATP released in wild type Syn7942 strains we had previously observed[19]. These engineering efforts resulted in the strain of *S. elongatus*, SynYLG6, that had the ability to secrete both glucose and ATP under defined conditions.

## Directed endosymbiosis to introduce carbon assimilation in yeast mutants via cyanobacterial endosymbionts

Next, we investigated if we could use our directed endosymbiosis approach to engineer SynYLG6 strains as endosymbionts within *S. cerevisiae cox2-60*. Particularly, in addition to using our previously developed polyethylene glycol (PEG)-induced fusion protocol, we designed selection conditions where the endosymbiotic cyanobacterial mutants SynYLG6 provide photosynthetically glucose and ATP to the yeast cells and, therefore, the yeast/cyanobacteria chimera are able to propagate under the defined photosynthetic conditions with bicarbonate-derived $CO_2$ as the carbon source (Fig. 2A). First, we introduced SynYLG6 within *S. cerevisiae cox2-60* using our PEG-induced fusion protocol[19]. Briefly, we generated *S. cerevisiae cox2-60* spheroplasts and mixed them with SynYLG6 cells. This mixture was added to a defined PEG buffer and was recovered under the defined selection conditions which favored the retention of the cyanobacterial endosymbionts within the yeast cells. Using this approach, we fused SynYLG6 to *S. cerevisiae cox2-60*. We also generated another control fusion between SynJEC3 (which lacked glucose secretion) and *S. cerevisiae cox2-60*. Fusion mixtures were recovered and selected by growing them on partial selection conditions containing a combination of non-fermentable carbon source, low levels of fermentable carbon source, and bicarbonate as a source of $CO_2$ (1% yeast extract, 2%

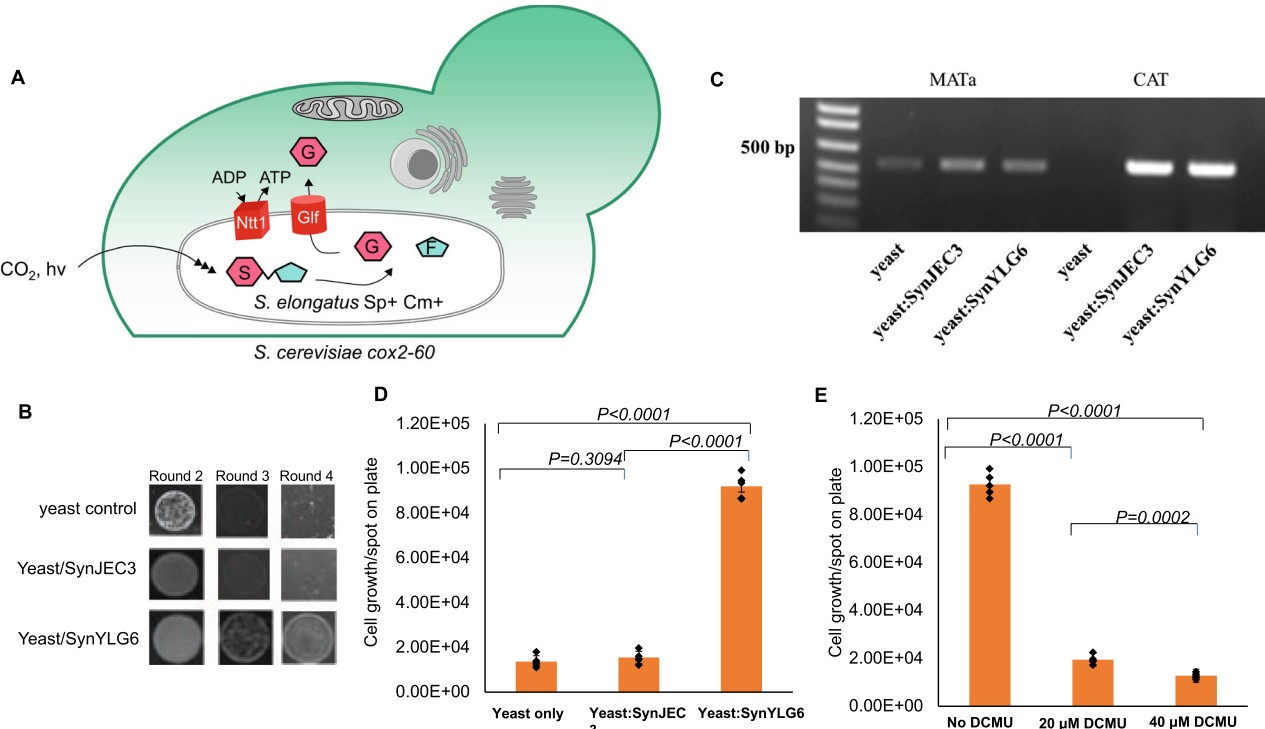

**Fig. 2 | *S. cerevisiae*–*SynYLG6* chimeras have a partially rescued respiration-competent phenotype. A** Our platform: We use suicide plasmid-based strategy to engineer cyanobacterial endosymbionts, *SynYLG6* strains, such that they can secrete glucose as well as ATP. *S. cerevisiae* mutants, deficient in ATP synthesis by oxidative phosphorylation under defined photosynthetic selection conditions, are used as the host strains. Engineered cyanobacteria strains, *SynYLG*, are then introduced into the yeast cells by a cell fusion process that is developed and optimized (see "Methods"). The yeast/cyanobacterial chimera are selected under defined photosynthetic selection conditions where the cyanobacterial endosymbionts provide glucose and ATP to the mutant *S. cerevisiae* host cells. Figure was prepared using BioRender. **B** Growth of *S. cerevisiae* cox2-60–*SynYLG6* chimeras on medium without extra carbon source. Additionally, no growth was observed in round 4 for yeast lacking intracellular *SynYLG6*. The experiment was repeated independently six times with similar results. **C** Total DNA isolated from spots grown on selection medium III contain the yeast-encoded *MATa* gene and

*SynYLG6/SynJEC3*-encoded chloramphenicol acetyltransferase (*CAT*) gene. The experiment was repeated independently six times with similar results. Complete data set is shown in Supplementary Fig. 4. **D** Growth trends of yeast-*SynYLG6* chimeras on Selection Medium III. Cells ($3.00 \times 10^3$) from round 3 were spotted on Selection Medium III and counted after 72 h growth. ($n$ = 5 technical replicates; data are presented as mean values +/− SEM.) *P*-values were calculated by two-tailed *t*-test comparing the two means. As listed the *P*-values for yeast and yeast:SynJEC3 are 0.3094, for yeast:SynJEC3 and yeast:SynYLG6 are <0.0001 and for yeast and yeast:SynYL6 are <0.0001. **E** Cell count analysis to demonstrate the effect of (3-(3,4-dichlorophenyl) −1,1-dimethylurea), DCMU, on *S. cerevisiae* cox2-60-SynYLG6 chimeras ($n$ = 5 technical replicates; data are presented as mean values +/− SEM). As listed the *P*-values for No DCMU and 20 μM DCMU treatment was <0.0001, for No DCMU and 40 μM DCMU was <0.0001 and 20 μM DCMU and 40 μM DCMU was 0.0002. Source data are provided as a Source Data file.

peptone, 1 M sorbitol, 3% glycerol, 0.1% glucose, 1X BG-11 containing 0.38 mM sodium bicarbonate; selection medium I). Recovered cells were propagated in 12 h light-dark cycles at 30 °C. After around 72 h, we observed small, distinct colonies for *S. cerevisiae* cox2-60-SynJEC3 and *S. cerevisiae* cox2-60-SynYLG6 fusions, but we did not observe yeast colonies with control *S. cerevisiae* cox2-60 cells. Multiple colonies were picked and re-plated for four consecutive rounds of regrowth: one round on selection medium II (1% yeast extract, 2% peptone, 1 M sorbitol, 3% glycerol, 0.1% glucose, 1X BG-11, 0.38 mM sodium bicarbonate, 50 mg/mL carbenicillin; selection medium II) and remaining rounds on selection medium III lacking both fermentable and non-fermentable carbon sources but containing bicarbonate as a carbon source (1% yeast extract, 2% peptone, 1 M sorbitol, 0.38 mM bicarbonate, 50 mg/mL carbenicillin, 1X BG-11; selection medium III) (Fig. 2B, 2D). Carbenicillin in selection mediums II and III allows us to eliminate any extracellular cyanobacteria. Consistent with the previously reported phenotype[19], the *S. cerevisiae* cox2-60 cells by themselves had low levels of growth on selection medium II and were unable to grow on selection medium III. We observed significant growth for *S. cerevisiae* cox2-60-SynJEC3 chimeras on selection medium II but minimal growth (if any) in subsequent rounds on stringent selection medium III. This is consistent with our design, where the SynJEC3 strains do not have the ability to

secrete glucose. On the other hand, the *S. cerevisiae* cox2-60-SynYLG6 chimeras were able to propagate in selection medium III, lacking both fermentable and non-fermentable carbon sources but containing bicarbonate as a carbon source, suggesting that the glucose and ATP provided by the cyanobacterial endosymbiont was sufficient to sustain the growth of the yeast/cyanobacteria chimeras under these conditions for multiple generations of growth (see Supplementary Table 1 for doublings). To demonstrate that photosynthesis was necessary for the chimera to survive, *S. cerevisiae* cox2-60/SynYLG6 were treated with (3-(3,4-dichlorophenyl) −1,1-dimethylurea), DCMU, an inhibitor of photosystem II[29]. We also observed loss of viability (Fig. 2E) indicating again the essential role of endosymbiont-mediated photosynthesis on the viability of the yeast/cyanobacteria chimeras. This suggested that upon fusion of the engineered cyanobacteria to the yeast cells, only the cyanobacterial mutants that were specifically engineered to provide glucose and ATP were able to recover the growth of the yeast mutants under the defined photosynthetic selection conditions containing bicarbonate as a carbon source. The above phenotypic reversion was indicative of metabolically active cyanobacterial endosymbionts within the yeast cells. To obtain more evidence for this, we isolated the total genomic DNA from the fused yeast cells that were propagated for multiple generations under selection growth conditions that

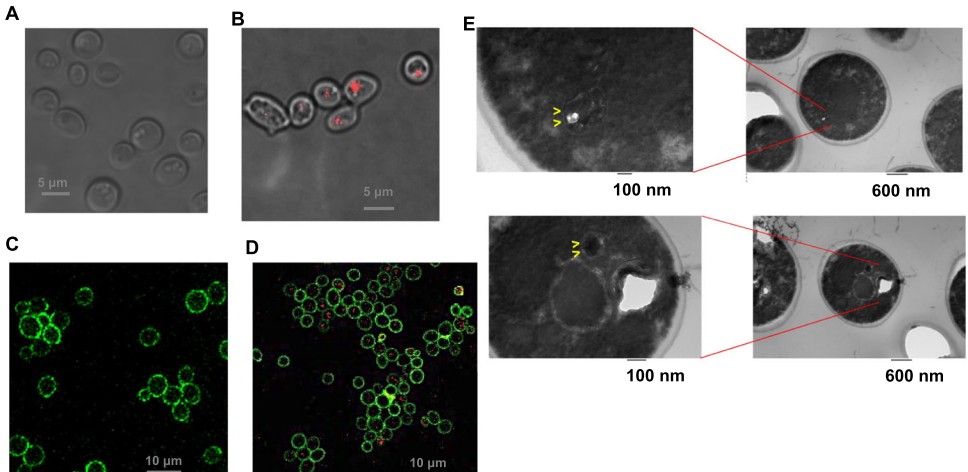

**Fig. 3 | Imaging intracellular cyanobacterial endosymbiont by fluorescent microscopy. A/B** pTIRF microscopic images of control yeast cells, and chimeric cells that were grown under the selection conditions (Ex. = 561 nm: Em. = 653/95 nm). Panels are merged images of pTIRF (red) and brightfield microscopy (gray). The experiment was repeated three times independently with similar results. **C/D** Fluorescence confocal microscopy images of the control yeast cells and the chimeric cells, which were grown under the selection conditions. The yeast cell wall was stained with Conn A-FITC (green, Ex. = 488 nm; Em. = 510/20 nm) and the presence of cyanobacteria was monitored by cyanobacterial fluorescence (red, Ex. = 561 nm; Em. = 650/20 nm). The experiment was repeated three times independently with similar results. **E** Samples imaged by transmission electron microscopy (TEM). Yellow arrows show characteristic cyanobacterial structures within the cytoplasm of the yeast cells. The experiment was repeated twice independently ($n = 2$) with similar results.

eliminated all the extracellular cyanobacteria, and then performed PCR analysis using oligonucleotides specific for either the yeast genome or the cyanobacterial genome. We detected the presence of both the yeast *MAT***a** gene and the cyanobacterial mutant chloramphenicol acetyltransferase (*CAT*) gene in the total genomic DNA (Fig. 2C and Supplementary Fig. 4), indicating the presence of both the yeast and the cyanobacterial genomes. The phenotypic growth studies, photosynthesis dependence and the genomic DNA analysis suggested the presence of engineered cyanobacterial endosymbionts capable of providing glucose and ATP to the yeast mutant cells.

### Characterization of cyanobacterial endosymbionts in yeast cells using microscopy

Next, we used microscopy analysis to determine the presence of cyanobacterial endosymbionts within the yeast cells. Cyanobacteria, including Syn7942-derived strains, have well-defined auto-fluorescent spectral properties due to the presence of chlorophylls and light-harvesting antenna proteins[30]. Therefore, we began by using fluorescence microscopy approaches for the characterization of the yeast/cyanobacteria chimeras. We first characterized our endosymbiotic chimeras using a home-built total internal reflection fluorescence (TIRF) microscope in the pTIRF mode to enable deep imaging inside live cells[19]. To obtain images, the yeast cells were first identified and imaged in brightfield mode and then pTIRF mode was used to check and detect cyanobacteria specific autofluorescence (excitation: 561 nm, emission: 661 nm)[31,32]. Under these conditions, no fluorescent signals were detected for the control *S. cerevisiae cox2-60* host cells (Fig. 3A). On the other hand, we observed distinct, punctate signals corresponding to cyanobacteria in the *S. cerevisiae cox2-60*-SynYLG6 chimeras that were propagated under selection conditions for multiple rounds of growth (Fig. 3B). This live cell imaging technique clearly detected the presence of cyanobacterial endosymbionts within the yeast cells propagated on bicarbonate as the carbon source.

While the live cell imaging allows us to detect the cyanobacterial signals within the yeast cells, we have previously developed confocal fluorescence microscopy approaches that allow us to measure the endosymbiont distribution in the fixed samples[19]. To generate samples for confocal fluorescence microscopy, we first propagated the *S. cerevisiae cox2-60*-SynYLG6 chimeras on iterative rounds of selection medium as described above and confirmed the detection of the cyanobacterial endosymbionts within the yeast cells by PCR analysis of the total genomic DNA and pTIRF microscopy. Once this was confirmed, the *S. cerevisiae cox2-60*- SynYLG6 cells were fixed with Karnovsky fixative[33], and stained with FITC-labeled concanavalin A (Conn A-FITC). Similarly, the control yeast cells, *S. cerevisiae cox2-60*, were also fixed and stained with Conn A-FITC. Samples were then analyzed by using a commercial Leica SP8 fluorescence confocal microscope according to the protocol we had previously developed[19] (excitation 488 nm and emission 525 nm corresponding to Conn A-FITC; excitation 561 nm and emission 661 nm for cyanobacterial autofluorescence) which allows us to detect both the yeast cells and the cyanobacterial cells simultaneously. As shown in Fig. 3C, D, we detected the presence of the cyanobacterial endosymbionts within the host yeast cells using confocal fluorescence microscopy further confirming the generation, selection, and propagation of the cyanobacterial endosymbionts within yeast cells after multiple rounds of growth under stringent selection conditions. No such signals were detected in the control yeast cells (Fig. 3C). To analyze the distribution of the cyanobacterial endosymbionts within the yeast cells over multiple rounds of propagation, we scanned a large number of cells using this approach. As shown in Supplementary Fig. 5, under our selection conditions, a significantly high fraction of yeast cells show signals specific to cyanobacteria as we propagate the chimeras on selection medium III. These experiments further characterized the presence and the distribution of the cyanobacterial endosymbionts within the yeast cells grown in bicarbonate as the carbon source.

To understand the localization (e.g., cytoplasmic vs vacuole trapped) of these endosymbionts within the host cells, we analyzed the yeast/cyanobacteria chimera using transmission electron microscopy (TEM). As before, we generated and propagated *S. cerevisiae cox2-60*-SynYLG6 chimeras under the selection conditions described above, harvested the cells, and fixed and treated them for TEM analysis. As shown in Fig. 3E, we observe characteristic structures corresponding to cyanobacteria within the yeast cells. Under these selection and propagation conditions, the cyanobacteria appeared to be localized within the cytoplasm and did not appear to be entrapped within any intracellular organelles.

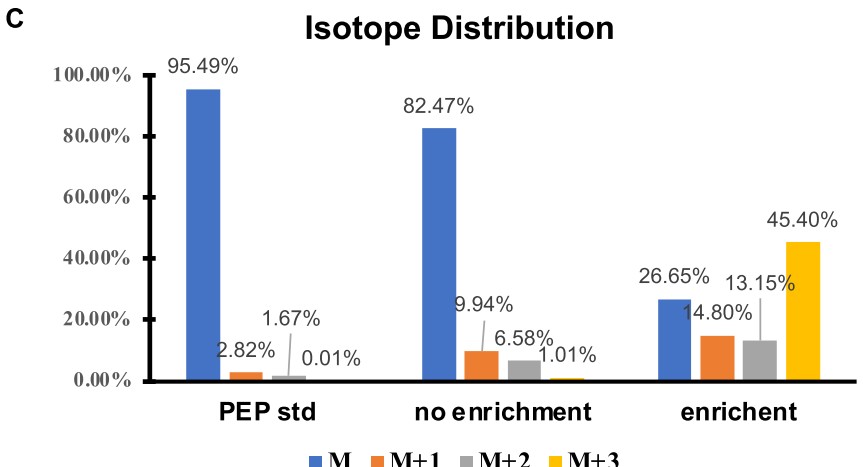

## Detection of assimilated carbon in yeast metabolism using metabolomics

Our next goal was to demonstrate that the $CO_2$ assimilated by the cyanobacterial endosymbionts gets incorporated into the yeast metabolism. Particularly, we investigated if the [13]C labeled bicarbonate would get incorporated into downstream metabolites in the yeast/cyanobacteria chimera (*S. cerevisiae* cox2-60–SynYLG6 chimeras) metabolome. To investigate this, we analyzed the glycolytic intermediate phosphoenol pyruvate (PEP) by using Liquid Chromatography Mass Spectrometry (LCMS) approach to investigate if the yeast/cyanobacteria chimera grown in presence of [13]C labeled bicarbonate would have isotopic enrichment due to assimilation followed by downstream metabolism (Fig. 4A). To start with, we first developed a PEP LCMS analysis method using PEP standards. Next, we developed methods to

**Fig. 4 | Metabolomics approach to detect isotope enrichment. A** The sources of inorganic carbon (C$_i$) utilized by air-grown Syn7942 are $^{13}$C-labeled HCO$_3^-$ in the culture medium and atmospheric CO$_2$. $^{13}$C-labeled HCO$_3^-$ is transported into the cell and is converted into CO$_2$ in the carboxysome, where it is subsequently converted into organic carbon via C$_3$ fixation. $^{13}$C-labeled sugars are exported by Syn7942 and act as yeast feedstock and are metabolized further. Ethanolic extracts of the yeast/cyanobacteria chimera metabolome are analyzed by LC/MS to search for yeast metabolites labeled with $^{13}$C originating from C$_i$. **B** Targeted analysis (*n* = 1) of unlabeled PEP (M + , blue) was measured at *m/z* 169, as well as $^{13}$C-labeled PEP isotopes of the precursor measured at three charged states: M + 1 (M + $^{13}$C, *m/z* 170, orange), M + 2 (M + 2$^{13}$C, *m/z* 171, gray), and M + 3 (M + 3$^{13}$C, *m/z* 172, yellow).

Visual representation of total ion chromatogram (TIC) scaled based on relative isotope distribution % of PEP and its charged states. Extracted ion chromatograms are included in Supplementary Fig. 6. **C** Isotope distribution in the PEP std. No $^{13}$C enrichment in PEP detected (1% M + 3) when *S. cerevisiae* cox2-60–*SynYLG6* chimeras were grown in unlabeled bicarbonate containing medium. $^{13}$C enrichment was detected in *S. cerevisiae* cox2-60–*SynYLG6* chimeras were grown in $^{13}$C labeled bicarbonate containing medium. Final results in Panel C are based data from extracted ion chromatograms included in Supplementary Fig. 6. Panel A was created with BioRender.com released under a Creative Commons Attribution-NonCommercial-NoDerivs 4.0 International license. Source data are provided as a Source Data file.

allow us to isolate metabolomes from yeast cells. Briefly, we generated yeast/cyanobacteria chimera for *S. cerevisiae* cox2-60–SynYLG6 that were propagated for three rounds either on selection conditions containing unlabeled bicarbonate or on selection conditions containing $^{13}$C labeled bicarbonate. We then isolated metabolomes from these cells and performed an analysis of PEP using ultra-high-performance liquid chromatography-MS (UHPLC-MS). Samples were injected (20 μL) into the Agilent 1290 Infinity II UHPLC system (Agilent Technologies, Santa Clara, CA) equipped with an Agilent Poroshell 120 EC-18 column (2.1 × 50 mm, 1.9 μm) with mobile phase A (0.1% formic acid in water) and mobile phase B (0.1% formic acid in acetonitrile). The flow rate was 0.6 mL/min. Using a 6500+ Triple Quadrupole MS (SCIEX, Redwood City, CA), mass spectra were acquired under positive ESI with the ion spray voltage of 5500 V. The source temperature was 350 °C. The curtain gas, ion source gas 1, and ion source gas 2 were 35, 65, and 60 pounds/square inch, respectively. Target *m/z* were 169, 170, 171, 172 for M, M + 1, M + 2, M + 3 correspondingly. Software Sciex Analyst 1.7.3 and MultiQuant 3.1 were used for data acquisition and analysis. Our data indicates that no $^{13}$C enrichment in PEP (1% M + 3) is detected when S. *cerevisiae cox2-60*–SynYLG6 chimeras were grown in an unlabeled bicarbonate-containing medium. On the other hand, significant $^{13}$C enrichment was detected in *S. cerevisiae cox2-60*–SynYLG6 chimeras when they were grown in $^{13}$C labeled bicarbonate-containing medium (Fig. 4B, C and Supplementary Fig. 6). Figure 4B. was created for visualization purposes of relative differences. Final results in Fig. 4C are based on data from extracted ion chromatograms in Supplementary Fig. 6.

## Metabolic engineering of the host yeast strains to introduce orthogonal metabolic pathways

Our next goal was to design a proof-of-concept study that demonstrates one possible application of this directed endosymbiosis approach, i.e., metabolic engineering of the photosynthetic yeast/cyanobacteria chimera to biosynthesize natural products. Significant advances have been made in the development of genetic tools in cyanobacteria and algae[34,35]. While photosynthetic platforms in cyanobacteria and microalgae have been previously explored, their metabolic potential is highly limited due to bottlenecks in CO$_2$ fixation, and in some cases due to the lack of molecular tools for metabolic engineering or due to the incompatibility of the enzymes from other organisms[36,37]. While expressing recombinant metabolic pathways is inherently complex, yeasts like *S. cerevisiae* have been engineered to produce many complex plant-derived molecules including terpenes, alkaloids and polyketides[38]. Extensive metabolic engineering efforts have been performed in model yeast strains to industrially produce high-value chemicals; for example high-value compounds like precursors of artemisinin are industrially produced in model yeast strains[39,40]. Yeast has also been engineered to encode orthogonal biosynthetic pathways (e.g., for monoterpenoid biosynthesis) that do not compete for important metabolic precursors[41–43]. All of this makes yeast a unique chassis for the metabolic engineering. However, these yeast-based approaches require significant feedstock carbon sources (e.g., glucose, ethanol) to drive the metabolic production of these chemicals.

We envisioned using directed endosymbiosis to couple the cyanobacterial photosynthetic to the vast biosynthetic and biocatalytic potential of yeast. Essentially, this approach would allow us to use the carbon-assimilating yeast/cyanobacteria chimeras to produce important terpenoids or their precursors in the absence of feedstock carbon sources like glucose or ethanol. To this end, we began by engineering an orthogonal metabolic pathway in our yeast host strains that would mediate the biosynthesis of a monoterpene, limonene[41]. We first made an integrative yeast DNA construct, pYG15, that allows us to incorporate foreign DNA into the defined sites in the yeast chromosome. Next, we cloned codon-optimized genes encoding *S.l.* NPPS1 (NP_001234633.1) from *Solanum lycopersicum* and *C.l.* LimS (Q8L5K3.1) from *Citrus x limon* into pYGL15 to generate DNA construct pYG16. In pYG16, the *S.l. NPPS1* gene expression was driven by Pgk-1 promoter and *C.l. limS* gene expression was driven by TPI-1 promoter. *S. cerevisiae* cox2-60 was transformed with pYG16 to generate a mutant *S. cerevisiae* cox2-60-limonene overexpression strain that contained genomic expression cassettes corresponding to *S.l.* NPPS1 and *C.l.* LimS (Fig. 5A-B, Supplementary Fig. 7). This strain was expected to divert the metabolic pools of dimethylallyl diphosphate (DMAPP) and isopentenyl diphosphate (IPP) to an orthogonal pathway that biosynthesizes neryl diphosphate (NPP) using *S.l.* NPPS1 enzyme. This is followed by the conversion of NPP to limonene by *C.l.* LimS enzyme. To investigate if this strain was capable of limonene production, we grew up this strain under fermentation conditions, performed solid phase microextraction (SPEM) followed by GCMS analysis. As shown in Fig. 5C-D, we clearly detected the production of limonene in this engineered host yeast strain. No signals corresponding to limonene production were detected in the control strain lacking the expression of *S.l. NPPS1* and *C.l. limS*.

## Directed endosymbiosis for biosynthesis of a monoterpene

Next, we investigated if we could couple the biocatalytic potential of yeast with the photosynthetic ability of cyanobacteria using directed endosymbiosis. We used *S. cerevisiae* cox2-60-limonene overexpression strain as the host strain and cyanobacterial mutant, SynYGL6 as the endosymbionts. Using directed endosymbiosis approach we described above, we generated *S. cerevisiae* cox2-60-limonene/SynYLG6 chimeras. Once the chimeras were generated, they were propagated under carbon assimilating, photosynthetic selection conditions in the presence of bicarbonate as the carbon source but in the absence of other feedstock carbon sources like glucose or glycerol. These chimeras were then transferred to 4 mL Liquid Selection Medium III in 20-mL glass vials with magnetic screws. Using a 2 cm-50/ 30 μm DVB/Carboxen™/PDMS StableFlex™ fiber, solid phase microextraction (SPME) was applied for extracting limonene from yeast/cyanobacteria chimera after 72 h of culturing. GCMS analysis was then performed on the extracted samples. Remarkably, under photosynthetic, selection conditions, we clearly detected the formation of limonene in the yeast/cyanobacteria chimera cultures (Fig. 5C, D). This proof-of-concept experiment clearly demonstrates the feasibility of using directed endosymbiosis to

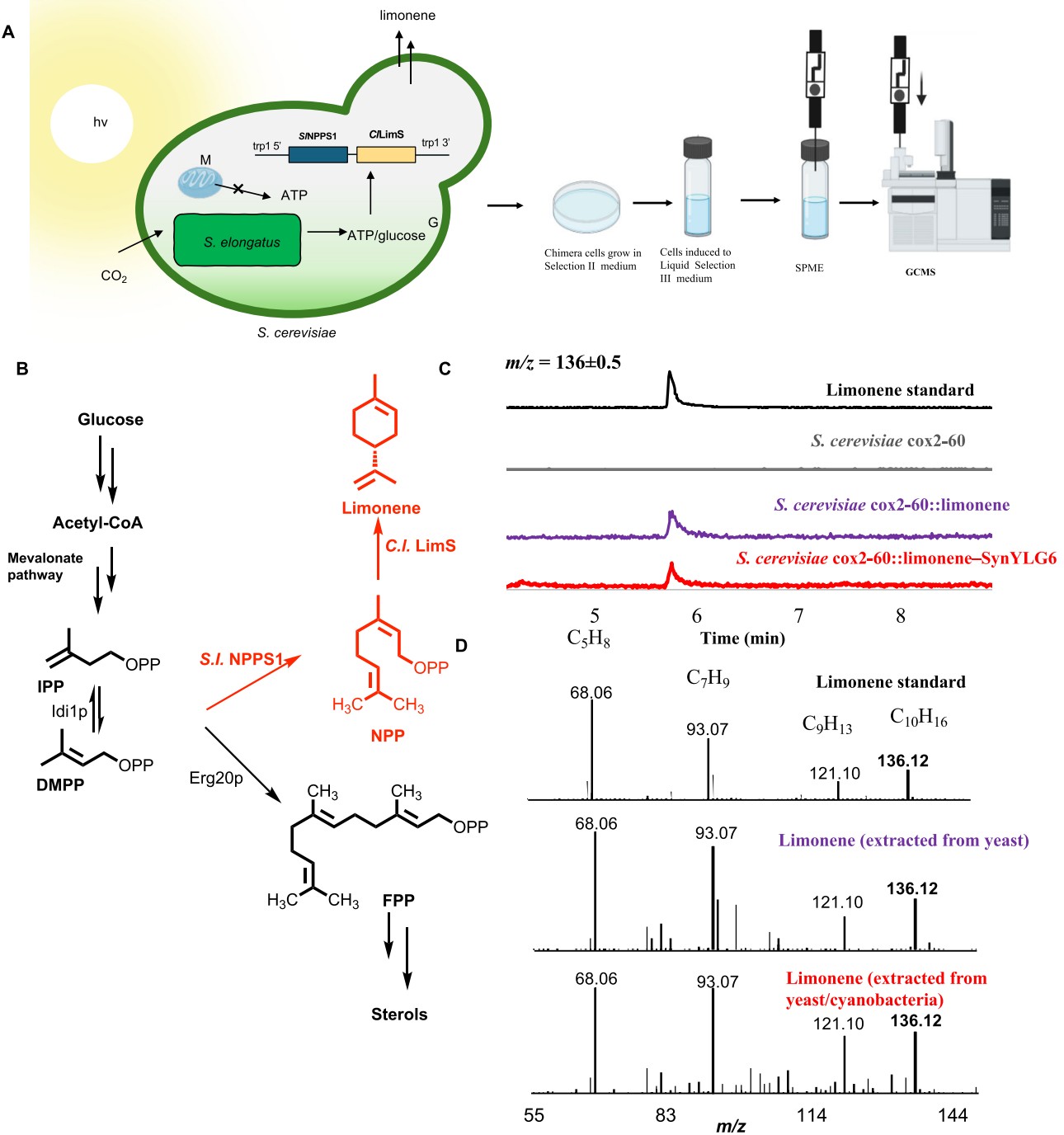

**Fig. 5 | Directed endosymbiosis for biosynthesis of a monoterpene. A** Our approach to coupling cyanobacterial photosynthesis to yeast metabolism using endosymbiosis to produce monoterpene, limonene. Figure was prepared using BioRender. **B** Engineering orthogonal pathway for limonene production using metabolic engineering. **C** GCMS experiments ($n = 1$) to detect limonene production. Extracted ion chromatograms ($m/z = 136 \pm 0.5$) corresponding to limonene to demonstrate the production of limonene in yeast/cyanobacteria chimera (red trace) and in engineered yeast strains possessing limonene biosynthetic pathway (purple trace) but not in control yeast samples lacking limonene biosynthetic pathway (gray trace). Black trace corresponds to commercially purchased limonene standard. **D** Mass spectra for limonene production: limonene standard, limonene extracted from engineered yeast strains and limonene extracted from engineered yeast/cyanobacteria chimera. Panel (**A**) was created with BioRender.com and released under a Creative Commons Attribution-NonCommercial-NoDerivs 4.0 International license. Source data are provided as a Source Data file.

biosynthesize natural products under photosynthetic selection conditions. This approach has the potential to be further optimized for photosynthetic bioproduction. Moreover, because the metabolic pathways are solely engineered in yeast, followed by rendering yeast photosynthetic using directed endosymbiosis, we anticipate that these platforms could be highly modular and genetically tractable for a wide range of metabolic engineering

efforts. The studies described in the section below, further expand the scope of this approach to any laboratory strain of yeast.

## Directed endosymbiosis to introducing carbon assimilation in standard laboratory strains of yeast

Generating mitochondrial mutants is technically challenging as compared to generating yeast nuclear genome mutants[44,45]. Therefore, we

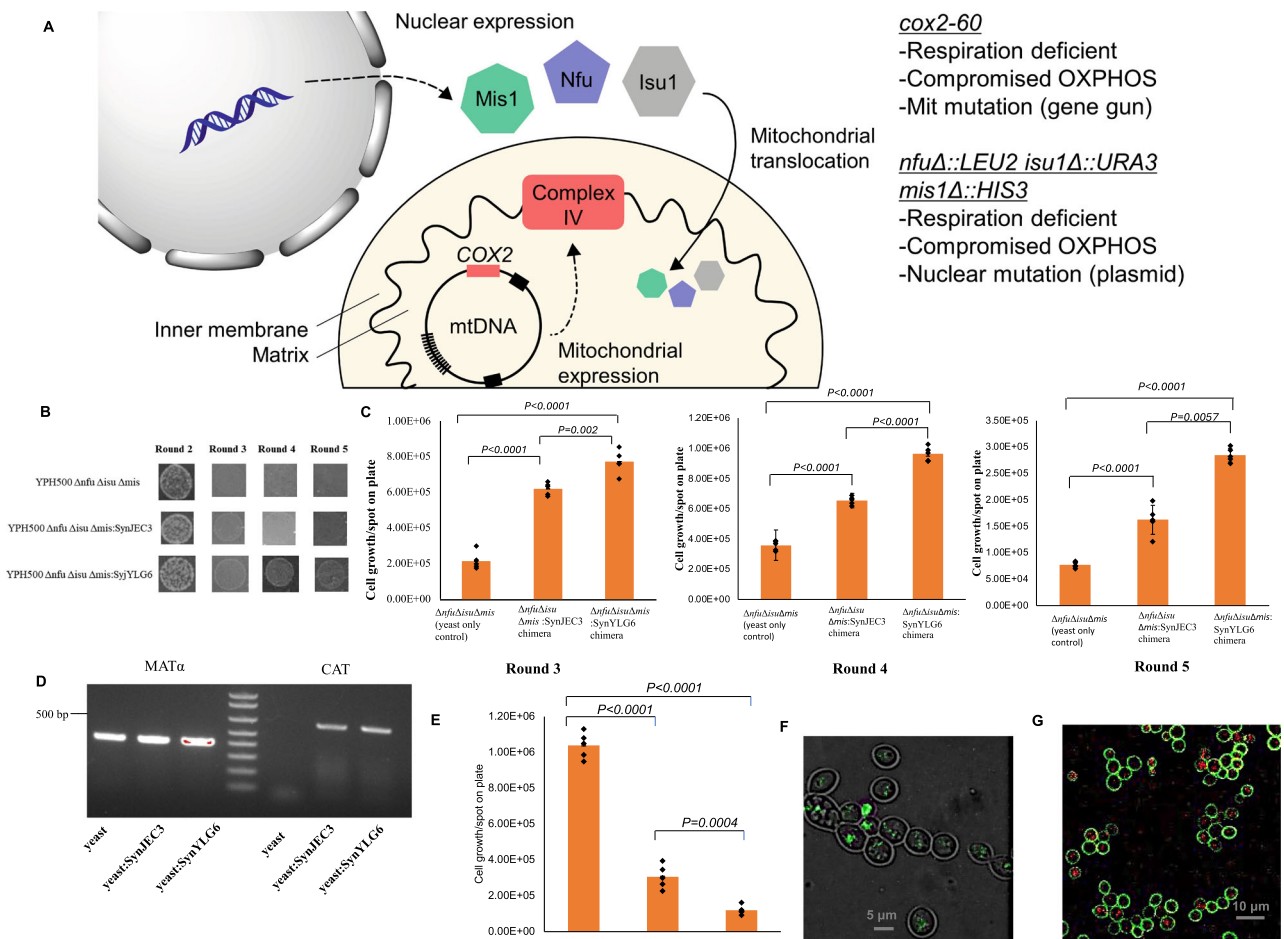

**Fig. 6 | Rescue of respiration deficient phenotype by YPH500 *nfuΔ::LEU2 isu1Δ::URA3 mis1Δ::HIS3* -SynYLG6 chimeras expressing a glf and invA protein.**
**A** Low levels of growth for YPH500 *nfuΔ::LEU2 isu1Δ::URA3 mis1Δ::HIS3* under glycerol as the carbon source. Nfu1: Protein involved in Fe-S cluster transfer to the mitochondrial proteins; Isu1: performs a scaffolding function during the assembly of iron-sulfur clusters; Mis1: Mitochondrial C1-tetrahydrofolate synthase. The figure was prepared using BioRender. **B** Growth of *S. cerevisiae* YPH500 *nfuΔ::LEU2 isu1Δ::URA3 mis1Δ::HIS3-SynYLG6*-chimeras on selection medium. The experiment was repeated three times independently with similar results and as listed the respective *P*-values are <0.0001, <0.0001 and 0.002. **C** Growth trends of *S. cerevisiae* YPH500 *nfuΔ::LEU2 isu1Δ::URA3 mis1Δ::HIS3* (yeast only strain), *S. cerevisiae* YPH500 *nfuΔ::LEU2 isu1Δ::URA3 mis1Δ::HIS3*-SynJEC3, *S. cerevisiae* YPH500 *nfuΔ::LEU2 isu1Δ::URA3 mis1Δ::HIS3*-SynYLG6 chimeras. Cells ($3.00 \times 10^3$) from each round were spotted on subsequent selection medium as described in the methods section and the final number of cells/spot on plate were determined after 48 h of growth (*n* = 5 technical replicates; data are presented as mean values +/− SEM). **D** Total DNA of

the yeast-*SynJEC3/SynYLG6* chimeras contains yeast *MATα* and *SynJEC3/SynYLG6 CAT* genes. The experiment was repeated three times independently with similar results. **E** Cell count analysis to demonstrate the effect of (3-(3,4-dichlorophenyl) −1,1-dimethylurea), DCMU, on *S. cerevisiae* YPH500 *nfuΔ::LEU2 isu1Δ::URA3 mis1Δ::HIS3*-SynYLG6 chimeras (*n* = 5 technical replicates; data are presented as mean values + /− SEM). *P*-values were calculated by a two-tailed *t*-test comparing the two means. As listed the *P*-values for No DCMU and 20 μM DCMU treatment was <0.0001, for No DCMU and 40 μM DCMU was <0.0001 and 20 μM DCMU and 40 μM DCMU was 0.0004. **F** pTIRF microscopic images of the chimeric cells that were grown under selection conditions (Ex. = 561 nm: Em. = 653/95 nm). Panels are merged images of pTIRF (green) and brightfield microscopy (gray). The experiment was repeated twice independently (*n* = 2) with similar results. **G** Fluorescence confocal microscopy images of the chimeric cells, which were grown under selection conditions. The experiment was repeated twice independently (*n* = 2) with similar results. Source data are provided as a Source Data file.

reasoned that in order to make this approach broadly applicable, we should develop a road map that would allow one to use our directed endosymbiosis approach, starting with standard laboratory strains of *S. cerevisiae* or strains of *S. cerevisiae* that have been engineered for synthetic biology applications. Particularly, we wanted to design an approach that allows us to start off with commercially available laboratory strains of yeast, make a few deletions in the yeast nuclear genomes and then use directed endosymbiosis that allows these yeast strains to propagate under photosynthetic growth conditions due the presence of cyanobacterial endosymbionts (Fig. 6A). In the approach described earlier, the host yeast cells, *S. cerevisiae cox2*-60, are mitochondrial mutants that are unable to prepare ATP under selection conditions. We envisioned that we could generate alternate host strains by engineering *S. cerevisiae* mutants with nuclear genome gene

deletions that have similar phenotype as mitochondrial mutant *S. cerevisiae cox2-60*[45,46] under the selection conditions (i.e., lack of mitochondrial ATP synthesis). Therefore, we targeted the nuclear genome encoded genes which are necessary for efficient mitochondria ATP synthesis. Previous studies demonstrate that the deletion of the nuclear genome encoded genes *nfu*, *isu1*, *mis1* results in the yeast mutants that are unable to grow on glycerol as a carbon source under respiration conditions due to compromised mitochondrial function resulting from the lack of efficient oxidative phosphorylation to generate ATP[47,48]. We started off with a commercially available standard laboratory strain of yeast YPH500 (ATCC # 76626) and sequentially generated following deletion mutants YPH500 *nfuΔ:LEU2:leu2*, YPH500 *nfuΔ::LEU2 isu1Δ::URA3*, YPH500 *nfuΔ::LEU2 isu1Δ::URA3 mis1Δ::HIS3*. We observed significant growth of YPH500 *nfuΔ::LEU2*

and YPH500 *nfuΔ::LEU2 isu1Δ::URA3* under respiration conditions with glycerol as the carbon source (2% glycerol, 2% yeast extraction, 2% peptide); on the other hand we observed significantly low levels of growth for YPH500 *nfuΔ::LEU2 isu1Δ::URA3 mis1Δ::HIS3* for multiple rounds of propagation under these conditions (Supplementary Fig. 8A). We further verified that YPH500 *nfuΔ::LEU2 isu1Δ::URA3 mis1Δ::HIS3* strain also had minimal growth when propagated in selection medium III (Supplementary Fig. 8B). This phenotype was similar to *S. cerevisiae cox2-60*, the host strain that we had used for the above-directed endosymbiosis studies.

Next, we evaluated whether the YPH500 *nfuΔ::LEU2 isu1Δ::URA3 mis1Δ::HIS3* cells were compatible for endosymbiosis with our cyanobacterial mutant SynYLG6. Using our fusion protocol, we fused SynJEC3 and SynYLG6 strains to YPH500 *nfuΔ::LEU2 isu1Δ::URA3 mis1Δ::HIS3* cells. We recovered these fusion mixtures on partial selection medium II as before, picked up individual colonies and plated them on selection medium II for one round of regrowth and on selection medium III for all subsequent rounds of regrowth. Under these selection conditions, we observed modest phenotypic recovery for YPH500 *nfuΔ::LEU2 isu1Δ::URA3 mis1Δ::HIS3*/SynJEC3 chimeras and an even higher phenotypic recovery for YPH500 *nfuΔ::LEU2 isu1Δ::URA3 mis1Δ::HIS3*/SynYLG6 chimera (Figs. 6B, C, see Supplementary Table 1 for number of doublings). To demonstrate that the endosymbiont photosynthesis was necessary for the chimeras to survive, YPH500 *nfuΔ::LEU2 isu1Δ::URA3 mis1Δ::HIS3/SynYLG6* chimeras were treated with DCMU, an inhibitor of photosystems. We also observed loss of viability (Fig. 6E) indicating again, the essential role of endosymbiont mediated photosynthesis on the viability of the yeast/cyanobacteria chimeras. This phenotypic recovery under photosynthetic selection conditions suggested the presence of cyanobacterial endosymbionts capable of providing ATP and glucose to the yeast cells under these conditions. In order to further verify the presence of cyanobacterial endosymbionts within yeast cells, we isolated the total genomic DNA from both YPH500 *nfuΔ::LEU2 isu1Δ::URA3 mis1Δ::HIS3*/SynJEC3 and YPH500 *nfuΔ::LEU2 isu1Δ::URA3 mis1Δ::HIS3*/SynYLG6 chimera, and PCR amplified the yeast *MATα* gene and the cyanobacterial mutant *CAT* gene in the total genomic DNA (Fig. 6D), indicating the presence of both the yeast and the cyanobacterial genomes. Further, we analyzed these chimeras by pTIRF and fluorescence confocal microscopy as before and detected the presence of cyanobacteria within yeast cells (Figs. 6F, G). The phenotypic growth studies, the genomic DNA analysis and the microscopy studies confirmed the presence of cyanobacterial endosymbionts capable of providing glucose and ATP to the YPH500 mutant strains. These studies provide a roadmap for transforming standard laboratory strains of yeast or strains of yeast that have been engineered for synthetic biology applications into photosynthetic yeast/cyanobacteria chimera.

## Discussion

There has been a long-standing quest to transform heterotrophic model organisms into partially or completely autotrophic organisms that are capable of assimilating carbon from $CO_2$ as a feedstock carbon source[1]. The predominant strategy that is used to accomplish this is using metabolic engineering to incorporate the components of the CCB cycle in model organisms, followed by adaptive evolution. In this study, we have developed a method that uses directed endosymbiosis to introduce carbon assimilation in model yeasts *S. cerevisiae*. Previously, we had developed a directed endosymbiosis approach that allows us to generate ATP providing cyanobacterial endosymbionts in the yeast cells[19]. In our current study, we investigated if we could engineer carbon-assimilating and sugar-secreting photosynthetic cyanobacterial endosymbionts (symbionts within the host cell) within the cytoplasm of the yeast cells to generate photosynthetic yeast/cyanobacteria chimeras that propagate in presence of $CO_2$ (or

bicarbonate) and do not require exogenous carbon sources like glycerol or glucose.

To accomplish this, we started off by engineering model cyanobacteria to secrete glucose assimilated from $CO_2$. Particularly, we performed metabolic engineering in cyanobacteria that results in the breakdown of sucrose into glucose and fructose, followed by cyanobacterial secretion of glucose. Having accomplished this, we further engineered these cyanobacteria to express proteins necessary for our directed endosymbiosis approach. Taken together, these engineering efforts resulted in the generation of a cyanobacterial mutant, SynYLG6, that was capable of (i) secreting glucose, (ii) secreting ATP in the presence of extracellular ADP and (iii) expressing SNARE-like proteins to improve the stability of cyanobacterial endosymbionts within host cells. Next, we developed fusion and selection conditions that allowed us to introduce SynYLG6 cells within the yeast mitochondrial mutants, *S. cerevisiae cox2-60*. Under these photosynthetic selection conditions, the cyanobacterial mutants provided ATP and glucose to the yeast mitochondrial mutants. These yeast/cyanobacteria chimeras were characterized by analyzing their viability under the photosynthetic selection conditions, analysis of the total genomic DNA, dependence on photosynthesis, pTIRF microscopy, fluorescence confocal microscopy and transmission electron microscopy. Further, we used metabolomics analysis to demonstrate that the assimilated carbon from cyanobacterial photosynthesis is incorporated into metabolites isolated from yeast/cyanobacteria chimeras.

As an application of this directed endosymbiosis approach, we demonstrated that orthogonal metabolic pathways can be engineered in the yeast/cyanobacteria chimeras to produce natural products under the photosynthetic selection conditions. While photosynthetic platforms in cyanobacteria and microalgae have been explored, their metabolic potential is highly limited[36]. On the other hand, *S. cerevisiae* has generally proved to be a highly versatile model organism that has been engineered to produce many complex plant-derived molecules, including terpenes, alkaloids and polyketides. These observations led us to investigate if we could couple the biocatalytic potential of yeast with the photosynthetic ability of cyanobacteria using directed endosymbiosis. We first performed metabolic engineering in our host yeast mutant strains to introduce orthogonal metabolic pathways for monoterpene, limonene, biosynthesis. Next, using directed endosymbiosis, we introduced the cyanobacteria within these engineered yeast strains to generate yeast/cyanobacteria chimeras. Remarkably, under the photosynthetic, selection conditions, we clearly detected the formation of limonene in the yeast/cyanobacteria chimeras. It is important to note that these metabolic pathways are solely engineered in yeast. This is followed by rendering these yeast strains to grow under photosynthetic using directed endosymbiosis. Because of the high genetic tractability of yeast strains and their compatibility with metabolic engineering, we anticipate that these approaches will be highly modular and genetically tractable for a wide range of metabolic engineering efforts. These proof-of-concept experiments demonstrate the feasibility of using a directed endosymbiosis approach for natural product biosynthesis. We anticipate that these platforms can be further optimized for photosynthetic bioproduction (i.e., conversion of $CO_2$ into chemicals).

To make this directed endosymbiosis approach broadly applicable, we also developed methods to transform standard laboratory strains of yeast or strains of yeast that have been engineered for synthetic biology applications into photosynthetic yeast/cyanobacteria chimeras. In our initial studies, the host cells we used were mitochondrial mutants, *S. cerevisiae cox2-60*. It is significantly challenging to edit the yeast mitochondrial genome as compared to the yeast nuclear genome. Therefore, we reasoned that in order to make this approach broadly applicable, we should design an approach that

allows us to start off with commercially available laboratory strains of yeasts, and make a few deletions in the yeast nuclear genomes to generate the host strains that are compatible for directed endosymbiosis. We demonstrated that we could perform standard yeast genetics on commercially available YPH500 strain of *S. cerevisiae*, to delete nuclear genes *nfu*, *isu1* and *mis1*. The resulting mutant strain was modestly compatible with our directed endosymbiosis approach. Notably, we saw growth enhancement under defined selection conditions for yeast/cyanobacteria chimeras (YPH500 *nfuΔ::LEU2 isu1-Δ::URA3 mis1Δ::HIS3* -SynYLG6) as compared to the yeast mutants (YPH500 *nfuΔ::LEU2 isu1Δ::URA3 mis1Δ::HIS3*) alone. These studies provide a roadmap for transforming standard laboratory strains of yeast or strains of yeast that have been engineered for synthetic biology applications into photosynthetic yeast/cyanobacteria chimera. We believe that such approaches could open up frontiers in biotechnology and sustainable synthetic biology[23] for $CO_2$ capture and $CO_2$ conversion into chemicals. In addition to this, given that these platforms are highly genetically tractable, they can be used to experimentally study the molecular details of the evolutionary adaptation of an additional endosymbiont/organelle in yeast. Studies on organelles and naturally existing endosymbionts have provided valuable insights into the evolutionary outcomes of endosymbiosis. Our current understating of these systems is mainly observational, as these systems are not genetically tractable. Therefore, there is a lack of understanding of the evolutionary trajectories, and the molecular drivers of the endosymbiotic transformation of bacteria into organelles is still poorly understood. Interestingly, despite the diversity of the endosymbiotic systems including the types of endosymbionts and the variety of host cells, some of the conserved molecular outcomes that serve as hallmarks of endosymbiosis include: (i) endosymbiont/organelle genome minimization and endosymbiont gene transfer, (ii) protein import/export systems, (iii) metabolic crosstalk between the endosymbiont and the host, (iv) possible modifications to the endosymbiont peptidoglycan membrane, and (v) host-controlled replication of endosymbionts/organelles. Due to the presence of these conserved features across various endosymbiotic systems, we believe that genetically tractable model endosymbiotic platforms that are able to synthetically reconstitute these outcomes could provide important insights into the molecular drivers and evolutionary trajectories of organelle evolution. Additionally, we think that comparing synthetic model systems to natural endosymbionts and organelles could provide a comprehensive view of organelle evolution.

## Methods

### Strains

*Synechococcus elongatus* strains were derived from *S. elongatus* PCC 7942 (Syn7942). This strain was donated from Prof. Susan Golden's lab (University of California San Diego, UCSD). *S. cerevisiae* ρ + (*MAT**a** leu2-3,112 lys2 ura3-52 his3ΔHindIII arg8Δ::URA3 [cox2-60]*)[46] was obtained from Schultz lab (Scripps Research) and *S. cerevisiae* YPH500 (*MATα ura3-52 lys2-801_amber ade2-101_ocher trp1-Δ63 his1-Δ200 leu2-Δ1*) generated in this study. These strains were used as hosts for *S. elongatus* endosymbionts. The *S. cerevisiae* YPH500 strain was purchased from ATCC.

### Growth media

*S. elongatus* cells were routinely cultured in BG-11 medium at 37 °C under 3000 lux. Yeast cells were routinely cultured at 30 °C in YPD medium (1% Bacto yeast extract, 2% Bacto peptone, 2% glucose) or synthetic defined (SD) medium (2% glucose, 0.67% yeast nitrogen base without amino acids) containing 50 mg/L carbenicillin.

All the fusion selection media are listed as follows:

Selection medium I: 1% yeast extract, 2% peptone, 0.1% glucose, 3% glycerol, 1 M sorbitol, 2% agar, 1X BG-11 salts containing 0.38 mM bicarbonate.

Selection medium II: 1% yeast extract, 2% peptone, 0.1% glucose, 3% glycerol, 1 M sorbitol, 2% agar, 1X BG-11 salts containing 0.38 mM bicarbonate, 50 µg/mL carbenicillin.

Selection medium III: 1% yeast extract, 2% peptone, 1 M sorbitol, 2% agar, 1X BG-11 salts containing 0.38 mM bicarbonate, 50 µg/mL carbenicillin.

NSII-recombinant *S. elongatus* mutant cultures were supplemented with 7.5 µg/mL chloramphenicol. NSI-recombinant *S. elongatus* mutant cultures were supplemented with 2 µg/mL spectinomycin and 2 µg/mL streptomycin. When noted, yeast medium was supplemented with adenine sulfate (Alfa Aesar A16964-09, 20 mg/L) or DCMU (A2B Chem AG00409).

### Construction of plasmids

All DNA oligonucleotide fragments were purchased from Integrated DNA Technologies (IDT). Gene-coding DNA fragments and plasmids were generated from the oligonucleotide either by PCR (Q5 Hot Start High-Fidelity 2X Master mix, NEB catalog # M0494S) or Gibson Assembly. Commercially synthesized DNA fragments (gBlock) sequences used in cloning are listed in Supplementary Data 1. Single-stranded oligonucleotide sequences are listed in Supplementary Data 2. Coding sequences were codon-optimized for *S. elongatus* expression using IDT codon optimization software (https://www.idtdna.com/CodonOpt). All reaction products were transformed into One Shot® *ccd*B Survival™ 2 T1ᴿ Chemically Competent Cells (Invitrogen A10460) according to the manufacturer's protocols. The plasmids pYG10 and pYG11 are derived from the Cyanovector pCV0063, pYG12, pYG13, pYG14, pYG15 and pYG16 are derived from the pAM199. Vector maps are included in Supplementary Fig. 1, and detailed vector map links are provided in Supplementary Table 2.

pYG10: pCV0063 was linearized by PCR using the oligonucleotides YG20/YG21. A gBlock of the *glf* and *invA* gene (from *Zymomonas mobilis*) codon optimized for *S. elongatus* was amplified by PCR using the oligonucleotides YG22/YG23. The amplified DNA fragment was inserted into linearized pCV0063 by Gibson Assembly to afford pYG10.

pYG11: pCV0063 was linearized by PCR using the oligonucleotides YG20/YG77. A gBlock of the *glf* codon optimized for *S. elongatus* was amplified by PCR using the oligonucleotides YG22/YG78. The amplified DNA fragment was inserted into linearized pCV0063 by Gibson Assembly to afford pYG11.

pYG12: pAM199 was linearized by PCR using the primers AM1608/YG64. A 1034-bp fragment of the 5′-end of the YPH500 *Nfu1p* gene (NC_001143.9) was amplified from YPH500 genomic DNA using the primers YG62/YG63. The amplicon was inserted into linearized pAM199 (*leu2* marker) by Gibson Assembly, and the construct was subsequently linearized using the primers YG65/AM1010. A 1053-bp fragment of the 3′-end of *Nfu1p* was amplified from YPH500 gDNA using the primers YG66/YG67, and the amplicon was inserted into the linearized vector by Gibson Assembly to afford pYG12.

pYG13: pAM199 was linearized by PCR using the primers YG252/YG254. A 1032-bp fragment of the 5′-end of the YPH500 *Isu1* gene (NC_001148.4) was amplified from YPH500 gDNA using the primers YG251/YG253. The amplicon was inserted into linearized pAM199 by Gibson Assembly, and the construct was subsequently linearized using the primers YG255/YG258. A 1024-bp fragment of the 3′-end of *Isu1* was amplified from YPH500 gDNA using the primers YG256/YG257 and the amplicon was inserted into the linearized vector by Gibson Assembly. Again, the construct was linearized using the primers YG259/YG262. The *ura3* maker was amplified from commercial plasmid pRS416 using the primers YG260/YG261, and the amplicon was inserted into the linearized vector by Gibson Assembly to finally afford pYG13.

pYG14: pAM199 was linearized by PCR using the primers YG116/YG118. A 1072-bp fragment of the 5′-end of the YPH500 *Mis1* gene (NC_001134.8) was amplified from YPH500 gDNA using the primers YG117/YG115. The amplicon was inserted into linearized pAM199 by

Gibson Assembly and the construct was subsequently linearized using the primers YG119/AM1010. A 1117-bp fragment of the 3′-end of *Mis1* was amplified from YPH500 gDNA using the primers YG120/YG121 and the amplicon was inserted into the linearized vector by Gibson Assembly. Again, the construct was linearized using the primers YG364/YG373. The *his3* maker was amplified from commercial plasmid pRS423 using the primers YG363/YG374 and the amplicon was inserted into the linearized vector by Gibson Assembly to finally afford pYG14.

pYG15: pAM199 was linearized by PCR using the primers YG697/YG699. A 998-bp fragment of the 5′-end of the *S. cerevisiae Trp1* gene was amplified from *S. cerevisiae* gDNA using the primers YG696/YG698. The amplicon was inserted into linearized pAM199 by Gibson Assembly and the construct was subsequently linearized using the primers YG701/YG703. A 982-bp fragment of the 3′-end of *Trp1* was amplified from *S. cerevisiae* gDNA using the primers YG700/YG702 and the amplicon was inserted into the linearized vector by Gibson Assembly to finally afford pYG15.

pYG16: pYG15 was linearized by PCR using the primers YG704/YG705. gBlock corresponding to the expression of *S.l. NPPS1* (NP_001234633.1) and *C.l. limS* (Q8L5K3.1) were codon optimized for *S. cerevisiae* and were inserted into linearized pYG15 by Gibson Assembly to afford pYG16.

pET/luciferase: pET28 was linearized by PCR using primer pairs BD1F/R. A 1693-bp fragment of the Luciferase gene was amplified from *E. coli* codon optimized gblock using primer pairs BD2F/R. The amplicon was inserted into a linearized pET vector by Gibson Assembly to afford pETLuciferase plasmid with kanamycin selectivity.

### Site-directed mutagenesis in cyanobacteria

Chromosomal integration of genes in the Syn7942 chromosome was achieved using a modification of the method developed by Golden and coworkers[25]. A total of 15 mL Syn7942 culture (OD730 = 0.5) was collected by centrifuging for 10 min at 3000 × *g*. The cells were washed once with 10 mL NaCl (10 mM) and then resuspended in 0.3 mL BG-11. To this suspension, the desired plasmid (4 μL) was added. The mixture was transferred to a 1.5 mL microcentrifuge tube the shaken in the dark (24 h at 30 rpm). Cells transformed mixture was spread on BG-11-agar medium supplemented with spectinomycin and streptomycin and incubated at 30 °C under 3000 lux for about 10 days. The quality of *S. elongatus* cultures was evaluated regularly by microscopy, streaking the cells onto BG-11-agar and by PCR analysis of recombinant loci. All mutant strains were generated by transforming corresponding recombination plasmids, for example, SynYLG3 was generated by transformation of Syn7942 with pYG10 to give a recombinant mutant which expresses a glycoside hydrolase and sugar transport from the NSI locus. SynYLG6 was generated by the transformation of SynYLG3 with pML17 to give a recombinant mutant, which ectopically expresses a ADP/ATP translocase from the NSII locus.

### Construction of *S. cerevisiae* genetic knockout strains

Plasmids are transformed to YPH500 as protocols described by Gietz et al.[49]. with some modifications: prepare a 5 mL overnight culture of the YPH500 in YPD liquid medium. On the day of transformation, inoculate ~200 μL of overnight culture into 20 mL YPD + carbenicillin. Incubate with shaking at 30 °C until OD600 = 0.6. Spin down 20 mL YPD culture 5k g for 5 min. Discard supernatant. Resuspend pellet in 1 mL sterile water. Spin down 5000 × *g* for 3 min and discard supernatant. Rinse again with 1 mL 0.1 mM LiOAc. Centrifuge 5000 × *g* for 3 min and discard supernatant. Add 300 μL 50% PEG 4000 directly onto cell pellet, add 50 μL 1 M LiOAc, 50 μL denatured single-stranded DNA (1 mg/mL Salmon sperm DNA), and plasmid. Pipette gently to mix. The tube sits at 30 °C for 30 min. Then transfer the tube to 42 °C water bath and heat shock for 1 hr. Invert the tube gently every 10–15 minutes

to ensure even heat distribution. Spin down on the tabletop centrifuge 3000 × *g* for 3 min. Discard supernatant and wash the pellet gently with water. Spin down 3000 × *g* for 3 min. Discard supernatant and resuspend the pellet in water and the plating on SD medium. The knockout strains were confirmed by growing in defined minimal medium without amino acid and genomic DNA analysis by PCR.

### Measurement of glucose release by cyanobacteria mutants expressing glycoside hydrolase and sugar transport

Syn7942 mutants—SynYLG3, SynYLG4, SynYLG5 and SynYLG6—were grown to reach OD730 = 0.12. For each assay, 20 mL cells were harvested by centrifugation (3000 × *g*, 5 min, room temperature), the supernatant was discarded, and the pellet was washed once with BG-11. Centrifuge the cells once more and resuspend in 3 mL BG-11 with 2 μg/mL spectinomycin and 2 μg/mL streptomycin as well as 150 mM NaCl. Add cultures to sterile glass tubes and shaken in the conventional light incubator as before. After 48 hours induction, remove 200 μL each culture to 1.5 mL Eppendorf tubes and centrifuge with 3000 × *g*, 5 min, the supernatant is collected for glucose test. The glucose concentration is measured using Glucose (GO) Assay Kit (GAGO20-1KT, Sigma, USA) according to manufacturer's protocol.

### Measurement of ATP efflux by cyanobacterial mutants expressing ADP/ATP translocase

The Luciferase enzyme was isolated after recombinant production from *E. coli* (BL21) harboring pETLuciferase plasmid after Isopropyl-beta-D-1-thiogalactopyranoside (IPTG) induction final concentration of 0.1 mM to produce N-(His)−6-tagged Luciferase fusion protein. The recombinant protein was isolated using standard protocol[50] using Ni-NTA affinity column and the fraction containing Luciferase enzyme was desalted using PD-10 column (GE healthcare, USA), finally stored in the storage buffer (10% glycerol, 50 mM HEPES, 100 mM NaCl, pH 8.0) at −80 °C for future ATP detection assay. The standard ATP detection protocol was followed[14].

### Introduction of mutant cyanobacteria to *S. cerevisiae* cells

We had previously developed methods to introduce cyanobacteria within yeast cells[19]. Syn7942 mutants were grown at 37 °C under constant light for 4 d. The cells (20 mL) were harvested (3000 × *g*, 10 min), washed twice with BG-11 and resuspended in BG-11 (300 μL). *S. cerevisiae* cells were grown in YPD medium (120 mL) until OD600 = 0.5. The yeast cultures were treated with Zymolyase to generate spheroplasts. The suspension was incubated for 1 h at 37 °C to form spheroplasts. The spheroplast suspension was mixed with cyanobacterial mutant strains and the fusion procedure was followed[19]. The recovered fusion mixtures were plated on Selection-I medium. After drying, a second layer of same medium was overlaid on the top and the plates were incubated at 30 °C for 3-4 days. The colonies were extracted from the plate, resuspended in 1 M sorbitol and spotted on Selection-II medium, incubated at 30 °C for 3 days. For subsequent rounds of propagation, cells were scraped from the surface of the plate, resuspended in 1 M sorbitol and spotted on Selection-III medium and incubated at 30 °C for 3 days per round of growth on selection medium. For fusions with YPH500 *nfuΔ::LEU2 isu1Δ::URA3 mis1Δ::HIS3* strains, colonies were extracted from the plate, resuspended in 1 M sorbitol and spotted on Selection-II medium, incubated at 30 °C for 2 days. For subsequent rounds of propagation, cells were scraped from the surface of the plate, resuspended in 1 M sorbitol and spotted on Selection-III medium and incubated at 30 °C for 2 days per round of growth on selection medium. Selection media conditions are crucial to maintain cyanobacterial endosymbionts within the host cells. This is particularly highlighted by the data shown in Supplementary Fig. 5 where we observe that as we increase the selection pressure from round 2 to round 3 (which has strict selection conditions, no glucose or glycerol)

we see significant levels of enrich of yeast cells containing cyanobacterial endosymbionts; particularly under strict selection conditions 40 to 60% of the yeast cells containing cyanobacteria.

## Cell count of yeast/cyanobacteria chimeras

Cell spots on Selection-III medium were collected from the agar plate, mounted on the glass slide and counted in triplicate using the Countess II FL Automated Cell Counter (Fisher cat. # AMQAF1000) following the manufacturer's instructions.

## Total genomic DNA isolation and PCR analysis

For qualitative PCR analysis, the total genomic DNA (gDNA) isolation was isolated using the Yeast DNA Extraction Kit (Thermo Fisher 78870) using manufacturer's protocol. Briefly, yeast cells were collected from the agar plates, centrifuged and resuspended with 300 µL Y-PER Reagent and incubated at 65 °C for 10 mins. This mixture was then centrifuged ($13,000 \times g$, 5 mins), the supernatant was discarded, and the pellet was resuspended with 200 µL DNA Releasing Reagent A and 200 µL DNA Releasing Reagent B, followed by incubation at 65 °C for 10 min. Protein Removal Reagent (96 µL) was added to this mixture and gently mixed. The mixture was then centrifuged ($13,000 \times g$, 5 mins) and the supernatant was transferred to another microcentrifuge tube and DNA was precipitated by adding 500 µL isopropyl alcohol. The mixture was centrifuged, and the pellet was washed with 400 µL 70% ethanol. A total of 50 µL water was added to the tube to afford the total gDNA.

## Analysis of yeast/cyanobacteria chimeras using pTIRF microscopy

Samples were prepared by collecting the cells from plates and washing them once with 1 M sorbitol. The samples were then analyzed using an in house TIRF microscopy set up we have previously developed[19]. The pTIRF images were acquired with a Photometric 512 Evolve EMCCD camera. Samples were imaged using a 100X oil immersion objective lens with NA = 1.4. The software for microscope control and data acquisition was developed using C + + and LabVIEW and all images were processed with ImageJ 1.53c.

## Analysis of yeast/cyanobacteria chimeras using fluorescence confocal microscopy

Samples were prepared by collecting the cells from plates with sterile water and washing with Hank's Buffered Salt Solution (HBSS: 140 mM NaCl, 1 mM $CaCl_2$, 5 mM KCl, 0.4 mM $MgSO_4$, 0.5 mM $MgCl_2$, 0.3 mM $Na_2PO_4$, 0.4 mM $KH_2PO_4$, 6 mM glucose, 4 mM $NaHCO_3$). The cells were resuspended in 50 µL HBSS containing ConA to a final concentration of 50 µg/mL. This mixture was incubated at 37 °C for 10 min, washed twice with HBSS, and incubated with Karnovsky fixative (2% glutaraldehyde, 2.5% paraformaldehyde). Commercial Leica SP8 fluorescence confocal microscope was used, and the samples were imaged through a 63X/1.40 HC PL APO Oil CS2 lens and excited with 488 nm and 561 nm laser light. The emission wavelengths 510/20 nm were detected with photomultiplier tube (PMT) detector, and emission wavelengths 650/20 range were detected using a high-sensitivity GaAsP HyD detector. All images were processed with ImageJ.

## Analysis of yeast/cyanobacteria chimeras using transmission electron microscopy

We had previously developed methods to image the yeast/cyanobacteria chimeras using transmission electron microscopy (TEM)[19]. Briefly, the cells were scraped with 500 µL water, centrifuged and resuspended in 20 µL Karnovsky fixative solution and kept at 30 °C for 1 hour. The cells were washed with 1 mL HBSS and the supernatant was aspirated, and the cells were resuspended with fixative buffer (2.5% EM-grade glutaraldehyde and 2.0% EM-grade formaldehyde in 0.1 M sodium cacodylate, pH 7.4) for 1 hour at room temperature. The

fixative mixture was centrifuged, washed twice with HBSS, and then resuspended with 1% osmium tetroxide and incubated for 90 minutes. Samples were subjected to 10-minute buffer rinse, and then resuspended in 1% aqueous uranyl acetate, followed by overnight incubation. The next day, samples were dehydrated via a graded ethanol series, culminating in propylene oxide. Following a graded propylene oxide; Epon812 series, the nuclear pellets were embedded in Epon812 before cutting. Ultrathin (ca. 90 nm) Epon sections on grids were stained with 1% aqueous uranyl acetate and lead citrate, and after the grids had dried, areas of interest were imaged at 160 kV, spot 3 using a Philips/FEI Tecnai G2 F20 S-TWIN transmission electron microscope.

## Yeast metabolite isolation and detection

To determine the $^{13}C$ labeled primary metabolites. The fusion cells *S. cerevisiae* cox2-60–SynYLG6 chimeras were spotted on Selection Medium III with (or without) $^{13}C$ carbonate. The cell pellets were harvested and washed twice by resuspension in 5 mL of tempered QS ((QS = 60% v/v methanol) and centrifugation at $4000 \times g$ and −20°C. A total of 4 mL of extraction solvent (=75% v/v ethanol, tempered to 85°C) was added to each pellet, the cells were suspended by vertexing, and the cells were extracted for 3 min at 85°C whereat vortexed a second time after 1.5 min of extraction and vortexed a third time after 3 min of extraction. The extracted pellets were rapidly cooled down but not frozen. The ethanolic extracts were separated from the cell debris by centrifugation at $4000 \times g$ and −20°C and decanting; they were pooled and evaporated to complete dryness in a vacuum centrifuge (Savant RVT400 from Thermo Scientific). Stored at −80°C when ready for detection. The pooled ethanolic extract was resuspended with LC-MS grade water. Insoluble particles were removed via centrifugation at $4000 \times g$ at 4°C for 10 min using a table centrifuge.

## LCMS methods to detect PEP

Analysis of PEP was conducted in yeast sample extracts using ultra-high-performance liquid chromatography-mass spectrometry (UHPLC-MS). Samples were injected (20 µL) into the Agilent 1290 Infinity II UHPLC system (Agilent Technologies, Santa Clara, CA) equipped with an Agilent Poroshell 120 EC-18 column ($2.1 \times 50$ mm, 1.9 µm) with mobile phase A (0.1% formic acid in water) and mobile phase B (0.1% formic acid in acetonitrile). The flow rate was 0.6 mL/min. Using a 6500+ Triple Quadrupole MS (SCIEX, Redwood City, CA), mass spectra were acquired under positive ESI with the ion spray voltage of 5500 V. The source temperature was 350 °C. The curtain gas, ion source gas 1, and ion source gas 2 were 35, 65, and 60 pounds/square inch, respectively. Target *m/z* were 169, 170, 171, 172 for M, M + 1, M + 2, M + 3 correspondingly (Supplementary Table 3). Software Sciex Analyst 1.7.3 and MultiQuant 3.1 were used for data acquisition and analysis.

## Limonene extraction from cultures

*S. cerevisiae* cox2-60 limonene overexpression strain was grown in YPD overnight at 30 °C and 150 rpm and subsequently induced in 4 mL liquid YPD using 20-mL glass vials with magnetic screw cap (Mikrolab Aarhus A/S). For fusion, fused cells *S. cerevisiae* cox2-60 limonene –SynYLG6 chimeras were spotted on Selection Medium II for 3 days, the cells were transferred to 4 mL Liquid Selection Medium III in 20-mL glass vials with magnetic screw. Solid phase microextraction (SPME) was applied to measure the limonene produced in yeast cells after 72 h of culturing, using a 2 cm-50/ 30 µm DVB/Carboxen™/PDMS StableFlex™ fiber followed by GC–MS analysis.

## GC-MS methods to detect limonene

GC–MS analysis was carried out using a ZB-5ms column and helium as a carrier gas with a constant velocity of 37 cm/s. Samples resulting from incubation of the SPME fiber for 30 min over the headspace of the yeast cultures were analyzed using the following temperature

program: initial temperature 60 °C, ramp to 105 °C with a rate of 3 °C min⁻¹, ramp to 240 °C with a rate of 20 °C min⁻¹, and hold for 5 min.

### Reporting summary

Further information on research design is available in the Nature Portfolio Reporting Summary linked to this article.

## Data availability

Data supporting the findings of this work are available within the paper and its Supplementary Information files. A reporting summary for this Article is available as a Supplementary Information file. Source data are provided in this paper.

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

## Acknowledgements

The research reported in this publication was supported by the National Institute of General Medical Sciences of the National Institutes of Health under Award Number R01GM139949. This funding was received by A.P.M. The content is solely the responsibility of the authors and does not necessarily represent the official views of the National Institutes of Health. We thank Dr. Xiuli Mao for her help with GCMS experiments.

## Author contributions

A.P.M., Y.G., J.E.C. and B.D. designed experiments. Y.G., J.E.C., and B.C.D. performed biochemical experiments, pTIRF microscopy experiments, and fluorescence confocal microscopy experiments. Y.G. and J.E.C. prepared samples for TEM imaging, and C.L.W. performed TEM imaging and analyzed TEM data. Y.G. isolated the metabolome and prepared samples for metabolomics, and A.V.U. and M.L.F. performed the LCMS experiments and analyzed the data for isotope enrichment. All wrote the manuscript.

## Competing interests

The authors declare that they have no competing interests.
