## [Peer Review File · Nature Communications]

Introducing carbon assimilation in yeasts using photosynthetic directed endosymbiosis.Editorial Note: This manuscript has been previously reviewed at another journal that is not operating a transparent peer review scheme. This document only contains reviewer comments and rebuttal letters for versions considered at Nature Communications.

Reviewers' Comments:

Reviewer #1:

Remarks to the Author:

The authors' responses are not entirely in line with my expectations. While I appreciate their efforts, I find myself disagreeing with most aspects of their responses. It seems to me that the authors may benefit from further insights into biological chemical production. For instance, while the authors highlight limitations in genetic and metabolic engineering tools for photosynthetic platforms, I believe that the primary bottleneck lies in CO₂ fixation rather than solely in molecular tools. Moreover, genetic engineering tools for photosynthetic organisms have seen considerable advancement, comparable to those for yeast and *E. coli*. While heterologous gene expression presents challenges across various organisms, including *E. coli* and yeast, it's important to acknowledge the complexity inherent in such processes.

Regarding the authors' assertion about the functional expression of eukaryotic cytochrome P450 enzymes, I'd like to point out that while expressing many P450s in prokaryotes can indeed be challenging, the potential of eukaryotic algae in this context should not be overlooked. Additionally, considering the cost-effectiveness of producing certain compounds from sugars derived from biomass rather than directly from CO₂, it's worth exploring alternative strategies for industrial production. The chemicals that should be derived directly from CO₂ are inexpensive and needed in large quantities, like biofuels and biopolymers. Almost all P450s show low activity, thus they should not be used for such chemicals.

Regarding the claim that this system could be utilized to study organelle evolution, I maintain some skepticism. While I acknowledge the potential insights that could be gained from studying the artificial system developed by the authors, I question how closely the observed phenomena mirror natural evolution. Further clarity on this point would enhance the manuscript's contribution to the field.

Lastly, considering the authors' previously reported method of constructing chimaeras, I believe that the manuscript lacks sufficient novel information, significance in results, and potential applicability for publication in this journal. Perhaps a journal specialized in synthetic biology would be a more suitable fit for this manuscript.

Reviewer #2:

Remarks to the Author:

The revised version of the paper is much more clear and I do not have any further comments on this very interesting paper.

Reviewer #3:

Remarks to the Author:

All my concerns have been addressed and I recommend publication.

Responses to reviewers:

Reviewer #1 (Remarks to the Author):

The authors' responses are not entirely in line with my expectations. While I appreciate their efforts, I find myself disagreeing with most aspects of their responses. It seems to me that the authors may benefit from further insights into biological chemical production. For instance, while the authors highlight limitations in genetic and metabolic engineering tools for photosynthetic platforms, I believe that the primary bottleneck lies in CO₂ fixation rather than solely in molecular tools. Moreover, genetic engineering tools for photosynthetic organisms have seen considerable advancement, comparable to those for yeast and *E. coli*. While heterologous gene expression presents challenges across various organisms, including *E. coli* and yeast, it's important to acknowledge the complexity inherent in such processes.

We agree and acknowledge these points. We have added these recommendations to the manuscript (page 12, lines 4-14).

Regarding the authors' assertion about the functional expression of eukaryotic cytochrome P450 enzymes, I'd like to point out that while expressing many P450s in prokaryotes can indeed be challenging, the potential of eukaryotic algae in this context should not be overlooked. Additionally, considering the cost-effectiveness of producing certain compounds from sugars derived from biomass rather than directly from CO₂, it's worth exploring alternative strategies for industrial production. The chemicals that should be derived directly from CO₂ are inexpensive and needed in large quantities, like biofuels and biopolymers. Almost all P450s show low activity, thus they should not be used for such chemicals.

We agree. We have removed the reference to P450s in the manuscript.

Regarding the claim that this system could be utilized to study organelle evolution, I maintain some skepticism. While I acknowledge the potential insights that could be gained from studying the artificial system developed by the authors, I question how closely the observed phenomena mirror natural evolution. Further clarity on this point would enhance the manuscript's contribution to the field.

We agree with the reviewer and acknowledge and clarify this (page 18, lines 14-29).

Lastly, considering the authors' previously reported method of constructing chimaeras, I believe that the manuscript lacks sufficient novel information, significance in results, and potential applicability for publication in this journal. Perhaps a journal specialized in synthetic biology would be a more suitable fit for this manuscript.

Our previous submission did not include any aspects of introducing carbon assimilation or metabolic engineering. We have now clarified this (page 6, lines 27-28 and page 16, lines 8-10).

We thank the reviewer #1 for evaluating our submission.

Reviewer #2 (Remarks to the Author):

The revised version of the paper is much more clear and I do not have any further comments on this very interesting paper.

We thank the reviewer #2 for their feedback that helped us to improve our submission.

Reviewer #3 (Remarks to the Author):

All my concerns have been addressed and I recommend publication.

We thank the reviewer #3 for their feedback that helped us to improve our submission.

Reviewers' Comments:

Reviewer #1:

Remarks to the Author:

Thank you for agreeing to all of my concerns. Since the authors agreed to all of my concerns, I must maintain my previous recommendation. I strongly believe this manuscript fits a journal specialized in synthetic biology.

Responses to reviewers:

Reviewer #1 (Remarks to the Author):

Reviewer #1 (Remarks to the Author):

Thank you for agreeing to all of my concerns. Since the authors agreed to all of my concerns, I must maintain my previous recommendation. I strongly believe this manuscript fits a journal specialized in synthetic biology.

We thank the reviewer #1 for their feedback.